# THE JPEG BLIND SPOT: EXPOSING A CRITICAL VULNERABILITY IN DOCUMENT TAMPERING DETECTION

## ABSTRACT

Deep models for document tampering detection increasingly rely on multimodal RGB+DCT architectures, implicitly assuming that JPEG block artifact grids (BAG) provide stable cross forgery cues. In this paper, we show that this assumption embeds a strong inductive bias that fails under minimal, adversarially constructed perturbations. Unlike natural images, where JPEG alignment is largely stochastic, document images contain sharply bounded glyph structures, making grid-aligned manipulations trivial for an adversary. We formalize this phenomenon through two complementary attacks. Grid-Aligned Forgery (GAF) preserves local JPEG block statistics by aligning copy move, splicing, or generative manipulations to the underlying 8×8 grid, removing the inconsistencies current models depend on. Pad–Recompress–Crop (PRC) globally shifts the JPEG grid while leaving RGB content unchanged, probing whether detectors meaningfully fuse RGB and DCT features or merely memorize position dependent frequency cues. To quantify these effects, we use two evaluation metrics, Attack Success Rate (ASR) for missing forged regions and False Positive Area (FPA) for unintended detections, which capture failure modes not measured by prior work. Evaluations on the DocTamper benchmark show that both attacks substantially degrade performance across a range of state-of-the-art and robustness-oriented (including adversarially robust) detectors, such as CAT-Net, DTD, FFDN, DocForgeNet, and ADCD-Net. Our findings indicate that many existing models exhibit a strong bias toward JPEG-grid statistics and highlight this as an opportunity for developing more robust multimodal architectures for real world, security critical document forensics.

## 1 INTRODUCTION

Document integrity is critical in high-stakes domains such as finance, government administration, and academia, where even minor data manipulations by malicious actors can lead to serious information security risks (Verdoliva, 2020). Meanwhile, the rapid progress and widespread availability of modern image editing technologies have made it increasingly convenient to create such forgeries, necessitating the development of efficient and robust methods for forgery detection (Nandanwar et al., 2021; Pun et al., 2023; Wu et al., 2019).

Recent deep learning (DL)-based detectors (Qu et al., 2023; Wang et al., 2022b; Riaz et al., 2025; Chen et al., 2025) have demonstrated strong performance on standard document tampering benchmarks (Qu et al., 2023; Wang et al., 2022b). However, many of these methods still largely rely on exploiting the frequency-domain artifact traces introduced by JPEG compression, particularly the discontinuities in the block artifact grids (Li et al., 2009), as discriminative cues for detecting manipulated regions in the image. While this strategy is well-motivated, given that JPEG is a widely adopted compression algorithm for storing images, we hypothesize that an over-reliance on frequency-domain traces introduces a critical vulnerability: instead of learning robust, semantically meaningful evidence of tampering, models may overfit to local grid statistics and fail whenever these statistics are preserved or systematically shifted. This reflects a problematic inductive bias driven by the training data distribution and by the structure of JPEG compression, which operates on non-overlapping $8 \times 8$ image patches.

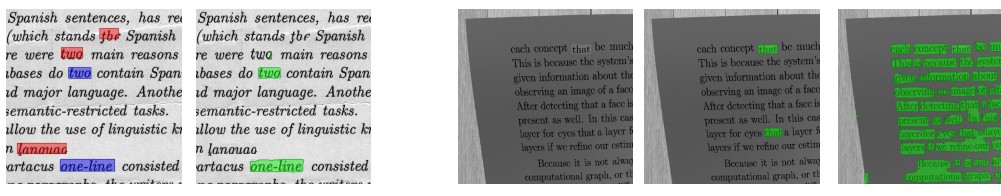

(a) GAF Attack          (b) PRC attack

Figure 1: Examples showing the effectiveness of our proposed forgery attacks. (a) The Grid-Aligned Forgery (GAF) attack (red) preserves local JPEG block statistics and successfully evades detection (green), in contrast to standard forgeries (blue). (b) The Pad–Recompress–Crop (PRC) attack (right) triggers a considerable amount of false positives by introducing only a subtle grid misalignment, as seen by the model's predictions before (middle) and after the shift (right).

To examine this failure mode, we introduce two complementary adversarial forgery procedures. (1) Grid-Aligned Forgery (GAF) aligns manipulated regions to the JPEG grid, preserving block-level DCT statistics. Despite the manipulated content remaining visually obvious to humans, grid preservation is often sufficient to bypass existing detectors. GAF generalizes across copy–move, splicing, and generative forgeries. (2) Pad–Recompress–Crop (PRC) shifts the JPEG grid by minimal padding and recompression, yielding out-of-distribution DCT patterns while keeping RGB content identical. PRC probes whether detectors have learned meaningful cross-modal correlations or merely memorized block level statistics. In principle, a model with robust global representations should remain invariant to such shifts, empirically, we show that this is not the case.

We evaluate our proposed methods on one of the largest available document tampering benchmarks, DocTamper Qu et al. (2023), and compare them against several state-of-the-art deep learning–based document-tampering detection methods. Across the DocTamper benchmark, both GAF and PRC induce significant failures in SotA detectors, reducing detection rates to as low as 1% and enabling systematic false positives. These results highlight a fundamental weakness in current architectures stemming from an over reliance on JPEG-grid statistics rather than learning robust, semantically grounded tampering cues. The main contributions of this work are following:

- We identify and formalize a critical inductive bias in JPEG-based forgery detectors arising from block-level DCT features.
- We introduce two complementary adversarial procedures, GAF and PRC, that exploit this bias through grid preservation and controlled grid shifts.
- We show that even minimal manipulations dramatically degrade SotA detectors, revealing fundamental limitations in current multimodal JPEG–RGB document tampering detection architectures.

## 2 RELATED WORK

**JPEG Forensics.** JPEG is the most widely adopted format for compressed images, and forensic analysis based on its artifacts has a long history. Early works focused on detecting double JPEG compression (Wang & Zhang, 2016; Fan & de Queiroz, 2003) and were later extended to tampering localization (Barni et al., 2010; Chen & Hsu, 2008; Li et al., 2009). For example, Barni et al. (2010) analyzed block-level statistics around suspected forgeries, while Chen & Hsu (2008) trained SVMs to discriminate forged from authentic regions. Other approaches modeled the probability of double compression at the DCT-block level Bianchi & Piva (2012) or extracted block artifact grids (BAGs) to localize tampering via grid discontinuities Li et al. (2009). For a comprehensive overview of classical approaches for JPEG-based forensics, see Verdoliva (2020).

**Deep Learning for Forgery Detection.** Deep learning shifted the field toward end-to-end detectors that combine RGB and frequency-domain or additional noise features. CNN-based approaches (Bayar & Stamm, 2018; Zhou et al., 2018; Amerini et al., 2017) and hybrid two-stream models (Kwon et al., 2021; Dong et al., 2022) demonstrated strong results on natural image tampering. More recently, attention-based and transformer-based architectures (Liu et al., 2022; Wang et al., 2022a) improved global reasoning but often lose sensitivity to subtle local artifacts. However, most of these methods remain optimized only for natural images, where manipulations are larger and visually

distinct, rather than document forgeries where edits are localized and text-like (Wu et al., 2019; Nandanwar et al., 2021).

**Deep Learning for Document Forgery Detection.** Since document forgeries are much more subtle compared to natural images, recent models explicitly introduce frequency-domain feature fusion strategies into deep neural networks for enhanced tampering detection. Abramova & Böhme (2016) proposed a method for detecting copy–move tampering in document images based on double quantization artifacts, however, this approach falls short when faced with multiple JPEG compressions. Wang et al. (2022b) introduced a two-stream Faster R-CNN (Ren et al., 2015) combining RGB and frequency features, but primarily targets SRNet-generated forgeries (Wu et al., 2019) rather than careful copy-paste tampering. For instance, Document Tampering Detector (DTD) Qu et al. (2023) is a recent state-of-the-art model a multi-modality Swin Transformer (Liu et al., 2021) model that employs a Frequency Perception Head (FPH) to capture tampering clues from DCT coefficients and a Multi-view Iterative Decoder (MID) to leverage multi-scale feature information from separate pixel-domain and frequency-domain input streams. FFDN (Chen et al., 2025) builds on the DTD (Qu et al., 2023) architecture by introducing a Vision Enhancement Module (VEM) and a Wavelet-like Frequency Enhancement (WFE) module for adaptive fusion of pixel-domain and frequency-domain features, and demonstrates state-of-the-art performance on multiple document tampering benchmarks. DocForgenet (Riaz et al., 2025) recently also propose to enhance feature fusion using dual-cross stream networks that fuse the frequency and pixel-level features via cross-attention. In addition, recent document-oriented models further extend multi-stream fusion. ADCD-Net (Wong et al., 2025) introduces an adaptive DCT weighting mechanism to handle block misalignment and employs hierarchical content disentanglement to reduce strong text–background bias, improving robustness under resizing and recompression. Similarly, the RTM baseline ASC-Former (Luo et al., 2025) leverages consistency-aware aggregation and gated cross-neighborhood attention to fuse RGB and transformed-domain cues, demonstrating strong performance on manually edited, highly concealed forgeries.

Despite several architectural advances, many current state-of-the-art document forgery detection models remain fundamentally dependent on block-level JPEG grid artifacts for identifying tampering. While this approach proves effective for natural image forgery, where random operations such as copy-move have only a 1/64 random chance of aligning with 8×8 JPEG block boundaries and thereby introduce detectable grid artifacts, document images present fundamentally different characteristics. In particular, the presence of discrete glyph structures with sharp foreground-background transitions makes it feasible to execute grid-aligned tamperings while maintaining visual plausibility. Our work is the first to systematically exploit this domain-specific vulnerability, demonstrating that current detection paradigms can be reliably circumvented and underscoring the critical need for more robust document tampering detection frameworks.

## 3 PRELIMINARIES

### 3.1 JPEG COMPRESSION MODEL

The encoding process of JPEG compression can be summarized in three main steps (1) The image is partitioned into $8 \times 8$ non-overlapping blocks, and a 2D discrete cosine transform (DCT) (Ahmed et al., 1974) is applied to each block independently to compute the DCT coefficients. (2) The resulting DCT coefficients are quantized using a quantization matrix $\mathbf{Q} \in \mathbb{N}^{8 \times 8}$, the values of which are determined according to the compression quality factor $f \in [0, 100]$. (3) Finally, the quantized DCT coefficients are entropy-coded (e.g., using Huffman and run-length encoding) in a lossless manner. Formally, given an original uncompressed image block $\mathbf{I}_{ij}$, JPEG compression followed by decompression with a quality factor $f$ can be expressed as

$$\mathbf{I}'_{ij} = \text{IDCT}(\mathcal{D}(\mathcal{Q}_f(\text{DCT}(\mathbf{I}_{ij})))) + \varepsilon, \tag{1}$$

where $\mathcal{Q}_f(\cdot)$ denotes quantization with $\mathbf{Q}$, $\mathcal{D}(\cdot)$ denotes the corresponding dequantization, and $\varepsilon$ accounts for rounding and truncation errors during decoding. In standard JPEG compression, Eq. 1 is applied to a single $8 \times 8$ block at position $(i, j)$ in the image and the full decompressed image is obtained by concatenating all independently reconstructed blocks of the image:

$$\mathcal{C}_f(\mathbf{I}) = \bigcup_{i,j} \mathbf{I}'_{ij}. \tag{2}$$

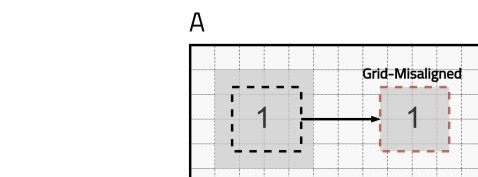 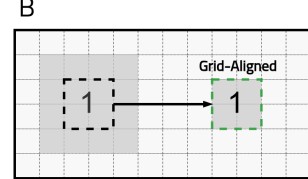

Figure 2: (a) A standard forgery disrupts the block artifact grid, leaving forensic traces of tampering. (b) Our proposed Grid-Aligned Forgery aligns the tampered text with the underlying $8 \times 8$ grid, in an attempt to preserve the per-block DCT statistics.

Since quantization is performed independently across each $8 \times 8$ block, horizontal and vertical discontinuities emerge at block boundaries, commonly referred to as block artifacts. In image forensics, the inconsistencies in block artifact grids between authentic and tampered regions provide strong cues for manipulation as illustrated in Fig. 2.

## 4 METHODOLOGY

Let $I_t \in \mathbb{R}^{3 \times H \times W}$ denote an input document image in RGB space. Then, for a standard image tampering setup (Li et al., 2009; Qu et al., 2023; Wang et al., 2022b), let $I_s$ be a source image from which the tampered content is obtained, together with a source bounding box $b_s = (x_s, y_s, w, h)$ specifying the position and size of the region to be copied. Let a corresponding target bounding box be $b_t = (x_t, y_t, w, h)$ specifying where this content is placed within the target image $I_t$. Then, let $\Pi$ be a unified forgery operator that crops the source region $b_s$ from $I_s$ and pastes it into the target region of $I_t$:

$$\Pi(I_t, I_s, b_s, b_t)_{:,i,j} = \begin{cases} I_{s:,\, i-y_t+y_s,\, j-x_t+x_s}, & x_t \leq j < x_t + w,\ y_t \leq i < y_t + h, \\ I_{t:,i,j}, & \text{otherwise.} \end{cases} \quad (3)$$

We consider three common types of forgeries in this work. For all types, the operator $\Pi$ remains identical; the only difference lies in how the source image $I_s$ is defined: **(1) Copy–move:** $I_s = I_t$, i.e., the source is the target image itself. **(2) Splicing:** $I_s \neq I_t$, i.e., the source is a different image from which the tampered region is extracted. **(3) Generative:** $I_s$ is produced by generative or rendering approaches. For details on how we perform splicing and generative forgeries, refer to Appendix A. Following previous works (Li et al., 2009; Qu et al., 2023; Wang et al., 2022b), we assume that after the forgery operation $\Pi$ is applied, the image again undergoes one or multiple JPEG compressions with a set of quality factors $F = \{f_1, f_2, \ldots, f_n\}$ and stored, resulting in the final tampered image $I'$:

$$I' = (\mathcal{C}_{f_n} \circ \cdots \circ \mathcal{C}_{f_1})(\Pi(I_t, I_s, b_s, b_t)) \quad (4)$$

Assuming a deep forgery detector $f_\theta : \mathbb{R}^{3 \times H \times W} \to [0, 1]^{H \times W}$ that outputs a tampering probability map $\hat{y} = f_\theta(I')$ over the forged image $I'$, the tampering operation can be modeled as a constrained adversarial attack (Zhou et al., 2022) that aims to minimize the detector's response over the desired tampered regions $b_t \in \mathcal{T}$:

$$\min_{b_s \in \mathcal{S},\, b_t \in \mathcal{T}} \sum_{b_t \in \mathcal{T}} \sum_{(i,j) \in \mathcal{P}(b_t)} \hat{y}_{i,j}, \quad (5)$$

where

$$\mathcal{P}(b_t) = \{(i,j) \mid x_t \leq j < x_t + w,\ y_t \leq i < y_t + h\}.$$

and $\mathcal{S}$ and $\mathcal{T}$ denote the desired candidate sets of source and target forgery regions, respectively. However, solving this optimization problem directly is intractable for two reasons. First, selecting appropriate candidate bounding boxes $(b_s, b_t)$ is nontrivial: the forger must identify semantically meaningful text regions $b_s$ from the source image $I_s$ that can be imperceptibly aligned with the target regions $b_t$ in RGB space. Since the coordinates and dimensions of these boxes can vary

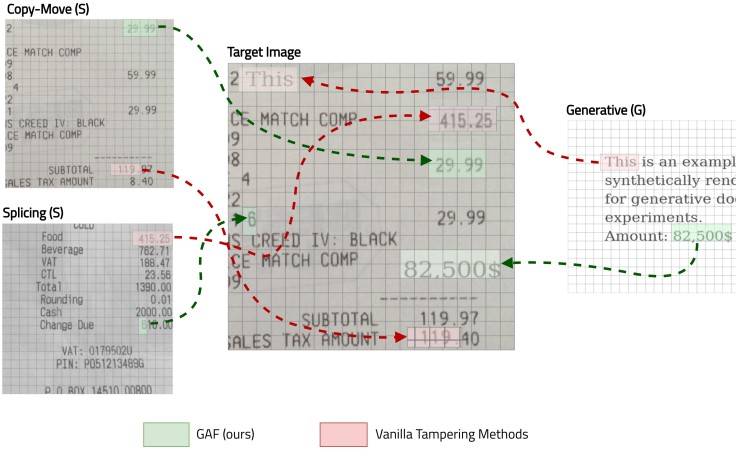

Figure 3: As shown, compared to standard tampering setups, GAF first aligns the source text boxes to the closest grid frontiers and then pastes them to target tampering locations also aligned with the grid of the target locations. Each cell of the grid is corresponds to a size of $8 \times 8$.

arbitrarily, this results in an exponentially large search space. Second, in realistic scenarios, the forger lacks white-box access to the detector $f_\theta$, making direct evaluation of $\hat{y}$ infeasible. While the first difficulty can be substantially mitigated using modern OCR tools, which we also employ to define the set of source candidates $\mathcal{S}$, the second limitation persists. To circumvent this, we propose tackling the problem indirectly by exploiting structural biases of modern detectors $f_\theta$, such as their over-reliance on frequency-domain DCT features for tampering detection.

### 4.1 ATTACK 1: GRID-ALIGNED FORGERY (GAF)

JPEG compression operates independently on non-overlapping $8 \times 8$ blocks, and modern tampering detectors (Qu et al., 2023; Riaz et al., 2025; Chen et al., 2025; Kwon et al., 2021) rely heavily on inconsistencies in block-level quantization artifacts as cues for manipulation. Building on this observation, we introduce Grid-Aligned Forgery (GAF), an adversarial forgery procedure that aligns manipulated regions exactly with the JPEG block structure to minimize detectable quantization mismatches (see Figure 3), with the complete pseudocode shown in Algorithm 1. For each OCR-detected box $b = (x, y, w, h)$, we align it to the JPEG grid using

$$\text{SNAP8}(b) = \big(8\lfloor x/8 \rfloor,\ 8\lfloor y/8 \rfloor,\ 8\lceil w/8 \rceil,\ 8\lceil h/8 \rceil\big),$$

which snaps the top-left corner to the nearest 8-pixel boundary and expands the width/height to the nearest grid-aligned size; only boxes with confidence above $\tau_{\text{conf}}$ are retained. Given a target region $b_t$, we select a source box $b_s$ by choosing the candidate whose area best matches that of $b_t$,

---

**Algorithm 1** Attack 1: Grid-Aligned Forgery (GAF)

---

**Require:** Target Image $I_t \in \mathbb{R}^{3 \times H \times W}$; Source Image $I_s \in \mathbb{R}^{3 \times H \times W}$; OCR bounding boxes $\mathcal{S}_{\text{OCR}} = \{(x_i, y_i, w_i, h_i, \text{conf}_i)\}_{i=1}^N$; JPEG quality factors $F = \{f_1, \dots, f_n\}$; target tampering box $b_t$; bounding box confidence threshold $\tau_{\text{conf}}$; size match threshold $\tau_{\text{area}}$
**Ensure:** Forged image $I'$, mask $M$
1: **fn** $\text{SNAP8}(b) := (8\lfloor x/8 \rfloor, 8\lfloor y/8 \rfloor, 8\lceil w/8 \rceil, 8\lceil h/8 \rceil)$, where $b = (x, y, w, h)$   ▷ Align box to 8-pixel grid
2: $\mathcal{S} \leftarrow \big\{ \text{SNAP8}(b) \mid (b, \text{conf}) \in \mathcal{S}_{\text{OCR}} \wedge \text{conf} \geq \tau_{\text{conf}} \big\}$   ▷ Filter OCR boxes by confidence and overlap
3: $\bar{b}_s \leftarrow \underset{\substack{\bar{b}_s \in \bar{\mathcal{S}} \\ \bar{b}_s \neq \bar{b}_t}}{\arg\min} |\text{area}(\bar{b}_s) - \text{area}(\bar{b}_t)|$   ▷ Select source box with similar size to target

   s.t. $\frac{\min(\text{area}(\bar{b}_s), \text{area}(\bar{b}_t))}{\max(\text{area}(\bar{b}_s), \text{area}(\bar{b}_t))} \geq 1 - \tau_{\text{area}},\ \text{IoU}(\bar{b}_s, \bar{b}_t) \leq \epsilon$
4: $I' \leftarrow (\mathcal{C}_{f_n} \circ \cdots \circ \mathcal{C}_{f_1})(\Pi(I_t, I_s, \bar{b}_s, \bar{b}_t))$   ▷ Copy–move patch from $\bar{b}_s$ to $\bar{b}_t$
5: $M \leftarrow \mathbf{0}_{3 \times H \times W}$; $M[y_t : y_t + h, x_t : x_t + w] \leftarrow 1$   ▷ Update mask for tampered region
6: **return** $I', M$

---

---

**Algorithm 2** Attack 2: Grid Shift via Pad–Recompress–Crop (PRC)

---

**Require:** Image $I \in \mathbb{R}^{3 \times H \times W}$; shift policy $\pi$ (fixed or random); JPEG qualities $\mathbf{q} = [q_1, \ldots, q_m]$; padding mode $\phi \in \{\text{edge}, \text{const}, \text{reflect}\}$
**Ensure:** Grid-shifted image $I'$
 1: $(\Delta x, \Delta y) \leftarrow \pi$, with $(\Delta x, \Delta y) \in \{0, \ldots, 7\}^2 \setminus \{(0,0)\}$
 2: $I_p \leftarrow \text{Pad } I$ with $(\Delta x, \Delta y)$ using mode $\phi$                                        ▷ Pad
 3: $I_c \leftarrow (\mathcal{C}_{f_n} \circ \cdots \circ \mathcal{C}_{f_1})(I_p)$                                      ▷ Recompress
 4: $I' \leftarrow I_c[\Delta y : \Delta y + H, \ \Delta x : \Delta x + W]$                                      ▷ Crop
 5: **return** $I'$

---

subject to an area-ratio constraint ($\geq 1 - \tau_{\text{area}}$) and low spatial overlap (IoU $\leq \epsilon$), exactly following Algorithm 1. The forgery operator $\Pi(I_t, I_s, b_s, b_t)$ copies content from $b_s$ into $b_t$, and a binary mask marks the manipulated region. To preserve block-wise artifact geometry (BAG), the manipulated image is then recompressed using the same JPEG quality factors $F = \{f_1, \ldots, f_n\}$ as the original acquisition pipeline. Algorithm 1 implements the copy–move variant (GAF-CM), while the same grid-alignment principle extends to splicing (GAF-S) and generative scenarios (GAF-G) by replacing the inserted content but always applying SNAP8 to maintain JPEG-block consistency (see Appendix A for more details). Overall, GAF produces visually plausible forgeries while significantly reducing the block-level inconsistencies exploited by state-of-the-art detectors, making it a strong adversarial baseline.

## 4.2 ATTACK 2: GRID SHIFT VIA PAD–RECOMPRESS–CROP (PRC)

While the GAF attack aims to minimize detector responses on target tampered regions, modern detectors' reliance on frequency-domain block artifacts suggests a complementary vulnerability. If these models discriminate forged from unaltered regions based on slight misalignments in the JPEG block grid, then deliberately introducing small global grid distortions should trigger the detector to classify many pixels as manipulated. Intuitively, this can be viewed as another type of adversarial attack that solves the inverse problem to Eq. 5: rather than minimizing detector responses, we seek to maximize the predicted tampering probability across the entire image. Formally, let $\Delta x, \Delta y$ define the grid shifts in horizontal and vertical directions, respectively, then Pad–Recompress–Crop ($\Pi_{PRC}$) operator for grid shift is defined as follows:

$$\Pi_{PRC}(I, \Delta x, \Delta y) = R_{\Delta x, \Delta y} \circ \mathcal{C}_q \circ P_{\Delta x, \Delta y}(I),$$

where $P_{\Delta x, \Delta y}$ pads the image on the left and top by $(\Delta x, \Delta y)$ pixels, and $R_{\Delta x, \Delta y}$ crops these pixels after JPEG recompression step described in Eq. 4. Applying this operator produces the attacked image

$$I' = \Pi_{PRC}(I, \Delta x, \Delta y) \tag{6}$$

The PRC attack is then formulated as an optimization over the grid shift $(\Delta x, \Delta y)$:

$$\max_{\Delta x, \Delta y} \sum_{(i,j) \in I'} f_\theta(I')_{i,j}, \quad \text{s.t. } (\Delta x, \Delta y) \neq (0,0),\ 0 \leq \Delta x \leq 7,\ 0 \leq \Delta y \leq 7. \tag{7}$$

By carefully selecting $(\Delta x, \Delta y)$, the Pad–Recompress–Crop (PRC) attack aims to exploits the model's sensitivity to grid misalignment, with the goal of producing as many false-positive tampering predictions as possible if the model is biased towards the frequency-domain features. Algorithm 2 provides the complete pseudocode for PRC attack.

## 5 EXPERIMENTS

### 5.1 EXPERIMENTAL SETUP

**Datasets.** We perform all evaluations on **DocTamper** (Qu et al., 2023), the largest publicly available dataset for document tampering detection. DocTamper provides 170k tampered English and Chinese document images created using copy–move, splicing, and generative methods. The dataset includes 120k training samples, a 30k primary test split (D-TestingSet), and two cross-domain test

splits: DocTamper-FCD (2k images from the Noisy Office dataset (Castro-Bleda et al., 2019)) and DocTamper-SCD (18k images from the HUAWEI Cloud dataset (Huawei Cloud, 2022)). All images are pre-forged and the dataset supplies pixel-level annotations of tampered regions. The different test sets exhibit substantial domain shift, allowing us to evaluate attack transferability under diverse conditions.

**Models.** We evaluate our attacks on six state-of-the-art forgery detectors that rely heavily on DCT-domain cues: CatNet (Kwon et al., 2021), DTD (Qu et al., 2023), DocForgeNet (Riaz et al., 2025), FFDN (Chen et al., 2025), RTM (Luo et al., 2025), and ADCD-Net (Wong et al., 2025). These models represent the strongest-performing systems on DocTamper and serve as canonical examples of block-artifact–driven detection pipelines. We apply both proposed attack families, Grid-Aligned Forgeries (GAF-CM, GAF-S, GAF-G) and Pad–Recompress–Crop (PRC), to assess their robustness.

**Evaluation Protocol.** Following the DocTamper protocol (Qu et al., 2023), we subject each test image to 1–3 JPEG recompressions with quality factors $\geq 75$ and the standard public seed. We report pixel-wise Precision (P), Recall (R), and F1-score (F) on all three test splits. To ensure a fair and controlled comparison when evaluating our attacks, we keep the underlying forgery type identical across all conditions. For the PRC attack, which does not introduce any new forgeries, we compare model performance directly against the unattacked setting (referred to as No Attack) on the original DocTamper forgeries (referred to as DocTamper). On the other hand, since the GAF attacks introduce additional forgeries into the dataset, we compare the degradation in model performance only on the forgeries created by our methods under two setups: without grid alignment (referred to as No Attack under each forgery type), and with GAF applied (denoted GAF-CM, GAF-S, and GAF-G for the copy-move, splicing, and generative cases respectively).

**Attack Evaluation Metrics.** For attack-specific evaluation, we define two additional metrics. To measure the overall effectiveness of the GAF attacks (GAF-CM, GAF-S, and GAF-G) in degrading the detector performance, we propose the attack success rate (ASR) metric[1]. We define $\text{ASR}_\tau$ as the total number of images on which the intersection-over-union (IoU) between the ground-truth and the predicted pixels is less than a target threshold $\tau$. We compute the ASR over multiple threshold levels $\tau \in \{0.0, ..., 0.5\}$ and report the average ASR over all thresholds. The intuition behind this is to quantify the tamperings bypassed by the detector under GAF. This indicates the effective concealment of the forgery from the detector. Similarly, to evaluate the effectiveness of the PRC attack in triggering false positives, we propose the False Positive Area (FPA) metric, which is computed as the fraction of pixels that are incorrectly predicted as tampered by the model. FPA measures the spatial extent of spurious tamper detections triggered by PRC, quantifying systemic false alarm generation. We report the mean FPA across all images, where higher values indicate that the attack successfully triggers more false positives.

**Implementation Details.** Because DocTamper provides only pre-forged images, we re-tamper them using GAF-CM, GAF-S, and GAF-G. We snap all source and target boxes to the JPEG $8 \times 8$ grid before applying the same recompression schedule. To obtain source and target boxes, we use EAST (Zhou et al., 2017) to detect text regions and then select size-matched box pairs while excluding regions overlapping with DocTamper's existing forgeries. This procedure preserves visual legibility and ensures that DCT-stream cues remain exploitable by the detectors. For PRC, we use the same evaluation metrics but substitute FPA for ASR to capture false-positive behavior.

## 5.2 Quantitative Evaluation: GAF and PRC

In Table 1, we present the quantitative evaluation results of our proposed attacks across all detectors and test splits. Following the evaluation protocol described in Section 5, we compare each model under the standard "No Attack" tampering setup against the three variants of Grid-Aligned Forgery (GAF-CM, GAF-S, GAF-G) and the Pad–Recompress–Crop (PRC) attacks. We summarize our key observations below.

**GAF-CM and GAF-S.** Across all three datasets, both GAF-CM and GAF-S induce substantial performance degradation, as measured by F1 score decline and elevated ASR/FPA rates. For instance,

---

[1]Note that this ASR metric is specific to forgery localization and differs from the ASR commonly used in adversarial robustness literature.

Table 1: Performance of state-of-the-art document tampering detectors under our proposed adversarial attacks. Results show severe performance degradation for most methods, including robustness-oriented models such as ADCD-Net, under the GAF-CM and GAF-S attacks, demonstrating systematic over-reliance on JPEG grid statistics. For GAF-G, the performance decline is milder but consistent.

| Detection Model | Attack Type | Forgery Type | TestingSet | | | | FCD | | | | SCD | | | |
|---|---|---|---|---|---|---|---|---|---|---|---|---|---|---|
| | | | P | R | F | ASR/FPA | P | R | F | ASR/FPA | P | R | F | ASR/FPA |
| CAT-Net | No Attack | DocTamper Original | 0.673 | 0.947 | 0.750 | - | 0.774 | 0.911 | 0.937 | - | 0.535 | 0.935 | 0.652 | - |
| | PRC | DocTamper Original | 0.544 | 0.731 | 0.624 | 0.011 | 0.568 | 0.604 | 0.595 | 0.042 | 0.437 | 0.726 | 0.546 | 0.009 |
| | No Attack | Copy-Move | 0.766 | 0.796 | 0.781 | 0.230 | 0.940 | 0.866 | 0.953 | 0.116 | 0.739 | 0.936 | 0.795 | 0.148 |
| | GAF-CM | Copy-Move | 0.744 | 0.550 | 0.633 | 0.570 | 0.810 | 0.415 | 0.549 | 0.890 | 0.695 | 0.524 | 0.594 | 0.508 |
| | No Attack | Generative | 0.871 | 0.736 | 0.798 | 0.153 | 0.911 | 0.939 | 0.874 | 0.007 | 0.869 | 0.698 | 0.774 | 0.083 |
| | GAF-G | Generative | 0.871 | 0.687 | 0.768 | 0.205 | 0.912 | 0.817 | 0.862 | 0.033 | 0.870 | 0.650 | 0.744 | 0.132 |
| | No Attack | Splicing | 0.828 | 0.946 | 0.937 | 0.151 | 0.939 | 0.942 | 0.941 | 0.114 | 0.804 | 0.824 | 0.814 | 0.112 |
| | GAF-S | Splicing | 0.826 | 0.669 | 0.739 | 0.348 | 0.237 | 0.102 | 0.142 | 0.754 | 0.776 | 0.580 | 0.664 | 0.369 |
| DTD | No Attack | DocTamper Original | 0.752 | 0.701 | 0.726 | - | 0.793 | 0.742 | 0.762 | - | 0.698 | 0.701 | 0.700 | - |
| | PRC | DocTamper Original | 0.057 | 0.823 | 0.107 | 0.177 | 0.094 | 0.215 | 0.121 | 0.094 | 0.057 | 0.749 | 0.105 | 0.111 |
| | No Attack | Copy-Move | 0.877 | 0.648 | 0.745 | 0.125 | 0.851 | 0.762 | 0.804 | 0.099 | 0.898 | 0.744 | 0.814 | 0.095 |
| | GAF-CM | Copy-Move | 0.446 | 0.298 | 0.357 | 0.512 | 0.304 | 0.029 | 0.054 | 0.845 | 0.686 | 0.381 | 0.490 | 0.439 |
| | No Attack | Generative | 0.953 | 0.479 | 0.638 | 0.143 | 0.948 | 0.782 | 0.857 | 0.045 | 0.942 | 0.470 | 0.627 | 0.172 |
| | GAF-G | Generative | 0.514 | 0.320 | 0.395 | 0.417 | 0.856 | 0.515 | 0.643 | 0.332 | 0.675 | 0.312 | 0.426 | 0.392 |
| | No Attack | Splicing | 0.928 | 0.752 | 0.831 | 0.068 | 0.839 | 0.737 | 0.785 | 0.101 | 0.933 | 0.791 | 0.856 | 0.057 |
| | GAF-S | Splicing | 0.572 | 0.472 | 0.517 | 0.310 | 0.237 | 0.102 | 0.142 | 0.754 | 0.727 | 0.499 | 0.592 | 0.276 |
| DocForgeNet | No Attack | DocTamper Original | 0.802 | 0.751 | 0.774 | - | 0.945 | 0.801 | 0.822 | - | 0.701 | 0.739 | 0.720 | - |
| | PRC | DocTamper Original | 0.050 | 0.874 | 0.095 | 0.215 | 0.067 | 0.263 | 0.106 | 0.131 | 0.049 | 0.810 | 0.091 | 0.141 |
| | No Attack | Copy-Move | 0.886 | 0.734 | 0.803 | 0.074 | 0.860 | 0.843 | 0.851 | 0.066 | 0.900 | 0.835 | 0.866 | 0.044 |
| | GAF-CM | Copy-Move | 0.396 | 0.293 | 0.333 | 0.532 | 0.182 | 0.017 | 0.032 | 0.879 | 0.578 | 0.367 | 0.449 | 0.452 |
| | No Attack | Generative | 0.955 | 0.550 | 0.698 | 0.089 | 0.946 | 0.790 | 0.861 | 0.028 | 0.943 | 0.540 | 0.687 | 0.107 |
| | GAF-G | Generative | 0.432 | 0.302 | 0.355 | 0.428 | 0.931 | 0.457 | 0.590 | 0.390 | 0.563 | 0.299 | 0.390 | 0.403 |
| | No Attack | Splicing | 0.930 | 0.821 | 0.873 | 0.045 | 0.859 | 0.831 | 0.845 | 0.062 | 0.932 | 0.854 | 0.891 | 0.033 |
| | GAF-S | Splicing | 0.495 | 0.429 | 0.459 | 0.353 | 0.163 | 0.104 | 0.127 | 0.762 | 0.593 | 0.469 | 0.524 | 0.317 |
| FFDN | No Attack | DocTamper Original | 0.873 | 0.940 | 0.956 | - | 0.927 | 0.905 | 0.916 | - | 0.805 | 0.819 | 0.812 | - |
| | PRC | DocTamper Original | 0.783 | 0.723 | 0.752 | 0.001 | 0.766 | 0.661 | 0.710 | 0.002 | 0.747 | 0.723 | 0.735 | 0.001 |
| | No Attack | Copy-Move | 0.830 | 0.801 | 0.815 | 0.112 | 0.888 | 0.943 | 0.915 | 0.024 | 0.864 | 0.889 | 0.876 | 0.062 |
| | GAF-CM | Copy-Move | 0.633 | 0.495 | 0.549 | 0.399 | 0.369 | 0.273 | 0.314 | 0.645 | 0.667 | 0.566 | 0.613 | 0.346 |
| | No Attack | Generative | 0.890 | 0.567 | 0.693 | 0.185 | 0.939 | 0.952 | 0.946 | 0.004 | 0.899 | 0.623 | 0.736 | 0.122 |
| | GAF-G | Generative | 0.871 | 0.483 | 0.621 | 0.257 | 0.932 | 0.904 | 0.918 | 0.027 | 0.866 | 0.520 | 0.650 | 0.218 |
| | No Attack | Splicing | 0.808 | 0.654 | 0.723 | 0.247 | 0.903 | 0.942 | 0.922 | 0.018 | 0.821 | 0.720 | 0.766 | 0.194 |
| | GAF-S | Splicing | 0.686 | 0.462 | 0.552 | 0.422 | 0.544 | 0.421 | 0.475 | 0.477 | 0.675 | 0.465 | 0.550 | 0.421 |
| RTM | No Attack | DocTamper Original | 0.745 | 0.701 | 0.722 | - | 0.794 | 0.699 | 0.739 | - | 0.643 | 0.682 | 0.662 | - |
| | PRC | DocTamper Original | 0.650 | 0.637 | 0.644 | 0.002 | 0.678 | 0.560 | 0.613 | 0.003 | 0.605 | 0.652 | 0.628 | 0.003 |
| | No Attack | Copy-Move | 0.674 | 0.641 | 0.657 | 0.251 | 0.702 | 0.693 | 0.698 | 0.228 | 0.712 | 0.757 | 0.734 | 0.173 |
| | GAF-CM | Copy-Move | 0.553 | 0.453 | 0.498 | 0.428 | 0.460 | 0.350 | 0.398 | 0.531 | 0.570 | 0.542 | 0.556 | 0.367 |
| | No Attack | Generative | 0.876 | 0.581 | 0.699 | 0.157 | 0.937 | 0.806 | 0.867 | 0.016 | 0.899 | 0.662 | 0.762 | 0.079 |
| | GAF-G | Generative | 0.959 | 0.522 | 0.649 | 0.201 | 0.929 | 0.755 | 0.933 | 0.028 | 0.879 | 0.592 | 0.708 | 0.126 |
| | No Attack | Splicing | 0.931 | 0.758 | 0.793 | 0.146 | 0.772 | 0.791 | 0.781 | 0.117 | 0.945 | 0.802 | 0.823 | 0.108 |
| | GAF-S | Splicing | 0.931 | 0.728 | 0.776 | 0.159 | 0.676 | 0.594 | 0.627 | 0.271 | 0.943 | 0.770 | 0.805 | 0.128 |
| ADCD-Net | No Attack | DocTamper Original | 0.789 | 0.823 | 0.806 | - | 0.866 | 0.770 | 0.815 | - | 0.690 | 0.799 | 0.740 | - |
| | PRC | DocTamper Original | 0.785 | 0.856 | 0.819 | 0.003 | 0.953 | 0.634 | 0.728 | 0.004 | 0.631 | 0.730 | 0.677 | 0.004 |
| | No Attack | Copy-Move | 0.815 | 0.498 | 0.618 | 0.154 | 0.930 | 0.638 | 0.757 | 0.115 | 0.828 | 0.604 | 0.698 | 0.221 |
| | GAF-CM | Copy-Move | 0.782 | 0.403 | 0.532 | 0.219 | 0.897 | 0.289 | 0.436 | 0.291 | 0.813 | 0.494 | 0.607 | 0.307 |
| | No Attack | Generative | 0.926 | 0.544 | 0.685 | 0.137 | 0.986 | 0.953 | 0.970 | 0.041 | 0.914 | 0.608 | 0.730 | 0.128 |
| | GAF-G | Generative | 0.903 | 0.424 | 0.577 | 0.211 | 0.989 | 0.896 | 0.940 | 0.005 | 0.889 | 0.426 | 0.575 | 0.249 |
| | No Attack | Splicing | 0.892 | 0.489 | 0.632 | 0.183 | 0.938 | 0.701 | 0.802 | 0.075 | 0.900 | 0.582 | 0.707 | 0.178 |
| | GAF-S | Splicing | 0.877 | 0.397 | 0.547 | 0.253 | 0.917 | 0.466 | 0.618 | 0.167 | 0.894 | 0.497 | 0.639 | 0.240 |

on the TestingSet (Copy-Move), CAT-Net shows an F1 reduction from 0.781 to 0.633 (15% relative decline), while the F1 on RTM degrades from 0.657 to 0.498 (24% decline). Notably, ADCD-Net, despite its adaptive DCT-weighting module explicitly designed for adversarial robustness, also demonstrates a considerable performance (F1) drop from 0.618 to 0.532 (14% decline). The vulnerabilities becomes more pronounced on the challenging FCD split, where the F1 for CAT-Net plummets from 0.953 to 0.549, for RTM from 0.698 to 0.398, and for ADCD-Net from 0.757 to 0.436. The ASR metric further corroborates the effectiveness of our attacks in a unified manner. CAT-Net's ASR increases from 0.230 (No Attack) to 0.570 under GAF-CM and from 0.151 to 0.348 under GAF-S, indicating that substantially larger portions of forged pixels evade detection entirely. DTD and DocForgeNet exhibit similar deterioration, with ASR values exceeding 0.50 across multiple splits. Even robustness-oriented architectures such as RTM and ADCD-Net demonstrate notable ASR increases (RTM: $0.251 \rightarrow 0.428$; ADCD-Net: $0.154 \rightarrow 0.219$ for Copy-Move). We provide additional results under varying ASR thresholds in Appendix C. Overall, our findings on GAF-CM/S attacks validate our central hypothesis that existing DCT-dependent architectures are fundamentally biased toward frequency-domain compression artifacts rather than semantic tampering cues, exposing critical limitations in current forgery detection paradigms.

**GAF-G.** In contrast to GAF-S/CM, GAF-G produces consistent but milder degradation across all models. This is expected as generative forgeries inherently disrupt JPEG history: the rendered text

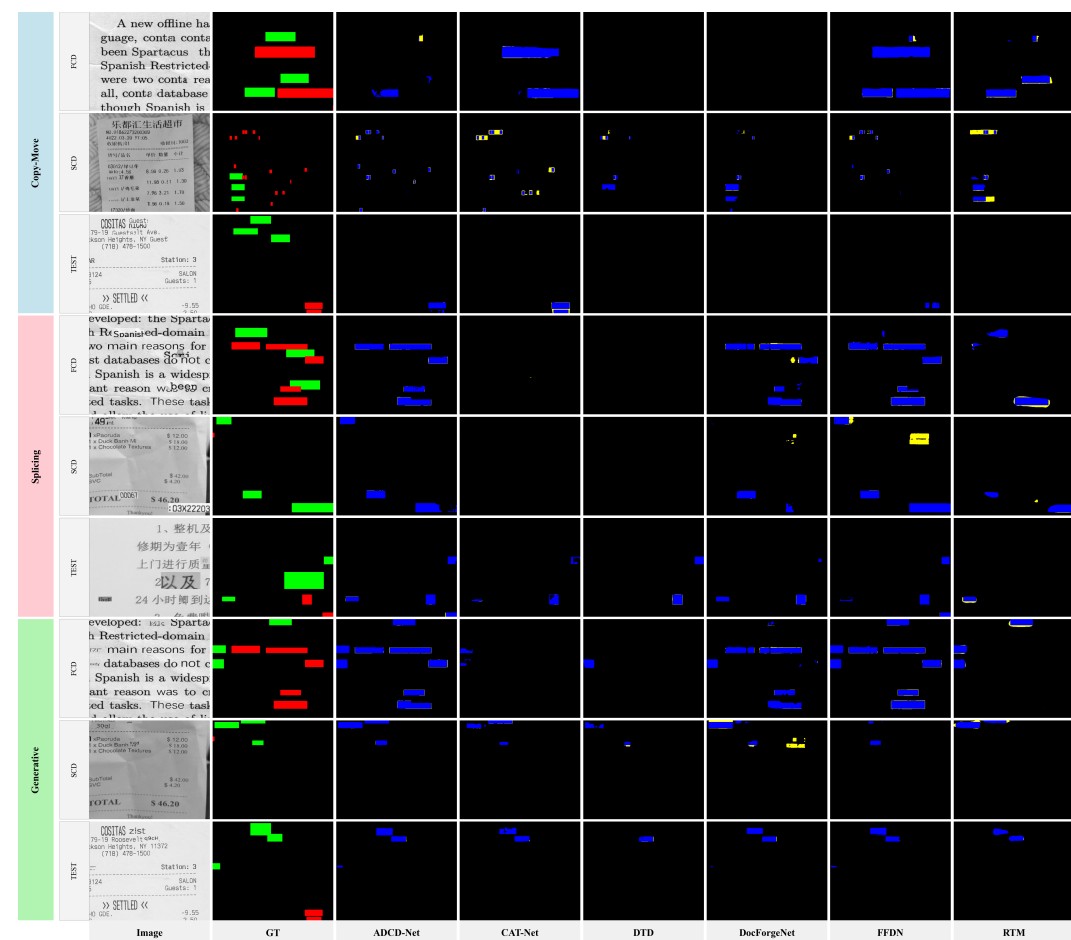

Figure 4: Qualitative comparison of the GAF attack across different state-of-the-art detection methods on the DocTamper dataset (Qu et al., 2023). Evidently, the GAFs (**Green**) evade detection at a much higher rate compared to the original tampering ground truth (**Red**), especially in the Copy-Move and Splicing scenarios, whereas they are less effective under Generative forgeries. **Blue** highlights True Positives (TP), and **Yellow** marks False Positives (FP).

patch carries mismatched quantization signatures, antialiasing patterns, and font-texture statistics that cannot be aligned with the host document, even after grid snapping. These signatures provide detectors with residual cues to partially localize tampering, resulting in moderate performance

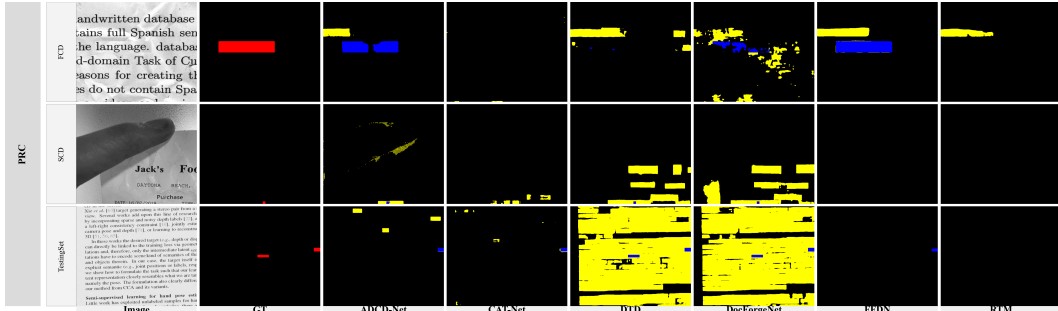

Figure 5: Qualitative comparison of the PRC forgery attack across different state-of-the-art detection methods on the DocTamper dataset (Qu et al., 2023). The PRC attack leads to severe failure cases, such as triggering a large number of false positives or causing complete failure to detect any forgery across existing models. **Red** denotes the original tampering ground truth. **Blue** highlights True Positives (TP), while **Yellow** marks False Positives (FP).

declines. For instance, CAT-Net's F1 drops from 0.798 to 0.768, DTD from 0.638 to 0.395, and DocForgeNet from 0.693 to 0.621, primarily driven by reduced recall. This behavior confirms that detectors rely on RGB inconsistencies only when generative content deviates from expected document statistics, while JPEG frequency artifacts remain their dominant decision signal.

**PRC.** PRC results demonstrate a characteristic failure pattern reflected in the FPA column: detectors such as DTD and DocForgeNet exhibit severe false-positive inflation (e.g., FPA 0.177 and 0.215 on TestingSet), often predicting large portions of clean text as tampered. Models like CAT-Net, FFDN, and RTM avoid this extreme behavior but still show clear F1 reductions driven by disrupted DCT alignment. For example, CAT-Net drops from 0.750 to 0.624, FFDN declines from 0.956 to 0.752, and RTM decreases from 0.722 to 0.644, while DTD and DocForgeNet collapse to 0.107 and 0.095. For FFDN specifically, we conduct an ablation study (see Appendix B) to investigate the source of its improved robustness. Our results suggest that FFDN's Vision Enhancement Module (VEM) enables more effective fusion of RGB and DCT features, which accounts for its superior performance under attack. Overall, the PRC results reveal that even minimally invasive, content-preserving grid shifts are sufficient to destabilize most detectors, either through large-scale over-detection or through reduced reliability, underscoring JPEG-grid dependence as a pervasive vulnerability.

## 5.3 QUALITATIVE ANALYSIS: GAF AND PRC

The qualitative results in Fig. 4 and Fig. 5 corroborate our quantitative findings. Under GAF-CM and GAF-S, DCT-reliant detectors (DTD, DocForgeNet) produce near-empty masks even for visually obvious forgeries once block-level inconsistencies are removed, indicating dependence on JPEG grid artifacts rather than visual evidence. FFDN demonstrates greater resilience through its broader feature extraction. Conversely, PRC exposes a complementary failure mode: globally shifting the JPEG grid triggers widespread false positives in DTD and DocForgeNet (consistent with high FPA in Table 1), incorrectly marking extensive regions as tampered despite unchanged RGB content. While CAT-Net, FFDN, and RTM avoid extreme false positives, they exhibit degraded masks. ADCD-Net shows more stability, though some false positives persist. These observations reinforce that current RGB+DCT detectors exhibit fragile dependence on JPEG grid statistics: when grids are perturbed, detectors either miss genuine manipulations or hallucinate tampering in clean regions.

## 6 LIMITATIONS

While our attacks demonstrate significant vulnerabilities in state-of-the-art forgery detectors, several limitations warrant discussion. GAF requires knowledge of the JPEG grid origin, an assumption shared with standard copy-move benchmarks that operate on uncropped images. When the grid position is unknown, PRC is explicitly designed for this scenario and does not require grid-origin information. Both attacks fundamentally rely on JPEG compression history. If a forgery is created entirely in lossless formats (e.g., PNG) without quantization structure, no exploitable JPEG grid exists and our methods are inapplicable. However, current RGB+DCT detectors are likewise not designed for such settings, and our focus remains on exposing vulnerabilities within JPEG-based forgery detectors, which represent the dominant paradigm in document forensics.

## 7 CONCLUSIONS

This work introduces two novel adversarial forgery attacks that exploit the over-reliance of state-of-the-art document forgery detectors on frequency-domain DCT features. Our experiments demonstrate that minor grid manipulations can severely mislead existing detection methods, leading to substantial performance degradation and systematic false positives. Our findings highlight critical safety and reliability concerns with direct implications for security and trust: malicious actors could selectively manipulate documents to evade detection, while the susceptibility to false positives could undermine the reliability of verification systems. By exposing these vulnerabilities, our research encourages the development of more robust document tampering detection systems that prioritize semantically grounded representations. In addition, our proposed forgery attacks can serve as a new form of evaluation benchmark for future research to audit the overall robustness of forgery detection models.

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
