## A   GAF-S and GAF-G Implementation Details

In this section, we provide the implementation details for the two tampering strategies in our Grid-Aligned Forgery (GAF) attack: GAF-G (Generative) and GAF-S (Splicing).

**GAF-G (Generative).** For the GAF-G attack, we perform text-level generative tampering in three steps. First, we detect a set of candidate text boxes in the image and select those that lie in pristine (non-tampered) regions based on the existing tamper mask. For each selected box, we estimate the local background color from surrounding pixels, remove the original text via inpainting, and then render a newly generated random string using a PIL-based text renderer. To ensure JPEG-consistent artifacts, each box is aligned to the JPEG grid: both the top-left coordinate and the box size are snapped to multiples of 8 pixels. Only the aligned region is then re-compressed using a randomly sampled JPEG quality between 75 and 95, producing realistic block-level DCT signatures. Finally, the tamper mask is updated by marking exactly the modified pixels as forged.

**GAF-S (Splicing).** For the GAF-S attack, we perform splicing by selecting a target text box from the current document and a source box from a different document. Both boxes are aligned to the JPEG $8 \times 8$ grid, ensuring that all boundaries match JPEG block edges. We remove the original content from the target region by filling it with an estimated background color and then paste the aligned source patch into this location. This produces a spliced image where the inserted content blends with the target background and remains consistent with JPEG compression structure. As in GAF-G, the tamper mask is updated to mark exactly the pasted (forged) area, while all other regions are left unchanged.

## B   Ablation study on FFDN Robustness

To assess whether the improved robustness of FFDN (Chen et al., 2025) against our proposed PRC attacks arises from its architectural enhancements, we conduct an ablation study focusing on its Wavelet-like Frequency Enhancement (WFE) module. FFDN builds upon the DTD (Qu et al., 2023) architecture, with the addition of Vision Enhancement Module (VEM) and WFE module to strengthen its feature representation and cross-modal integration. By removing WFE, FFDN can be isolated into a configuration that effectively corresponds to a DTD base model with an added VEM for enhanced RGB–DCT feature fusion. Therefore, we evaluated FFDN under two sets of conditions: (1) FFDN w/ WFE and (2) FFDN w/o WFE, and the results are reported in Table 2. As shown, removing WFE has a negligible effect on the predictive performance of the model, both in the absence and presence of forgery attacks. This strongly suggests that FFDN's VEM is solely responsible for the additional robustness of FFDN against the PRC attacks and therefore its VEM could be considered as one viable solution against the PRC attacks. However, it is worth mentioning that while the addition of VEM allows FFDN to not trigger false positives, its performance still degrades significantly with F1-scores dropping by 12.4%, 22.95%, and 9.7% on TestingSet, FCD, and SCD respectively under PRC attacks. This suggests that PRC can still serve as a general robustness test for evaluating model resilience to distribution shifts, even when models are not explicitly vulnerable to grid-alignment biases that trigger false positives.

Table 2: Performance comparison on FFDN (Chen et al., 2025) model with and without the WFE module in FFDN. Results show that the WFE module FFDN model has a negligible effect on the model predictions.

| Detection Model | Attack Type | Forgery Type | TestingSet | | | | FCD | | | | SCD | | | |
|---|---|---|---|---|---|---|---|---|---|---|---|---|---|---|
| | | | P | R | F | ASR/FPA | P | R | F | ASR/FPA | P | R | F | ASR/FPA |
| FFDN w/ WFE | No Attack | DocTamper Original | 0.873 | 0.940 | 0.956 | - | 0.927 | 0.905 | 0.916 | - | 0.805 | 0.819 | 0.812 | - |
| | PRC | DocTamper Original | 0.783 | 0.723 | 0.752 | 0.001 | 0.766 | 0.661 | 0.710 | 0.002 | 0.747 | 0.723 | 0.735 | 0.001 |
| | No Attack | Copy-Move | 0.830 | 0.801 | 0.815 | 0.112 | 0.888 | 0.943 | 0.915 | 0.024 | 0.864 | 0.889 | 0.876 | 0.062 |
| | GAF-CM | Copy-Move | 0.633 | 0.495 | 0.549 | 0.399 | 0.369 | 0.273 | 0.314 | 0.645 | 0.667 | 0.566 | 0.613 | 0.346 |
| | No Attack | Generative | 0.890 | 0.567 | 0.693 | 0.185 | 0.939 | 0.952 | 0.946 | 0.004 | 0.899 | 0.623 | 0.736 | 0.122 |
| | GAF-G | Generative | 0.871 | 0.483 | 0.621 | 0.257 | 0.932 | 0.904 | 0.918 | 0.027 | 0.866 | 0.520 | 0.650 | 0.218 |
| | No Attack | Splicing | 0.808 | 0.654 | 0.723 | 0.247 | 0.903 | 0.942 | 0.922 | 0.018 | 0.821 | 0.720 | 0.766 | 0.194 |
| | GAF-S | Splicing | 0.686 | 0.462 | 0.552 | 0.422 | 0.544 | 0.421 | 0.475 | 0.477 | 0.675 | 0.465 | 0.550 | 0.421 |
| FFDN w/o WFE | No Attack | DocTamper Original | 0.875 | 0.933 | 0.953 | - | 0.930 | 0.900 | 0.915 | - | 0.813 | 0.802 | 0.808 | - |
| | PRC | DocTamper Original | 0.794 | 0.714 | 0.747 | 0.001 | 0.759 | 0.659 | 0.705 | 0.002 | 0.755 | 0.705 | 0.729 | 0.001 |
| | GUCM | Copy-Move | 0.827 | 0.791 | 0.809 | 0.118 | 0.893 | 0.939 | 0.915 | 0.024 | 0.863 | 0.873 | 0.868 | 0.074 |
| | GAF-CM | Copy-Move | 0.618 | 0.471 | 0.534 | 0.417 | 0.360 | 0.262 | 0.303 | 0.657 | 0.648 | 0.547 | 0.593 | 0.369 |
| | GUG | Generative | 0.889 | 0.557 | 0.695 | 0.193 | 0.941 | 0.948 | 0.944 | 0.004 | 0.898 | 0.606 | 0.724 | 0.137 |
| | GAF-G | Generative | 0.868 | 0.470 | 0.609 | 0.272 | 0.932 | 0.890 | 0.910 | 0.033 | 0.860 | 0.500 | 0.632 | 0.240 |
| | GUS | Splicing | 0.780 | 0.610 | 0.695 | 0.288 | 0.908 | 0.937 | 0.922 | 0.020 | 0.795 | 0.681 | 0.734 | 0.228 |
| | GAF-S | Splicing | 0.650 | 0.422 | 0.511 | 0.465 | 0.539 | 0.412 | 0.467 | 0.482 | 0.638 | 0.428 | 0.512 | 0.462 |

# C    ASR PERFORMANCE ACROSS IOU THRESHOLDS

In Figure 6, we present Attack Success Rate (ASR) results across varying IOU thresholds ($\tau$) for all evaluated models under GAF-S, GAF-G, and GAF-CM attacks. Several key observations emerge from this analysis. First, it is evident that the performance decline under GAF-CM attack increases consistently as the IOU threshold becomes more lenient (higher $\tau$ values). This behavior is observed across all models, with one notable exception: ADCD-Net demonstrates substantially higher robustness, maintaining ASR $\approx 0$ at thresholds $\tau \leq 0.1$ and achieving the lowest ASR values even at $\tau = 0.5$ across all three datasets. Overall, CAT-Net, DTD, and DocForgeNet exhibit the highest vulnerability to this attack, showing similar degradation patterns across all datasets.

Under GAF-S attack, models exhibit largely similar behavior to GAF-CM, with a notable shift in relative robustness: RTM consistently demonstrates the strongest resistance in this scenario, outperforming ADCD-Net. This suggests that the splicing-based manipulation strategy exposes different architectural weaknesses compared to copy-move operations, with RTM's design being particularly effective at handling splicing-based manipulations.

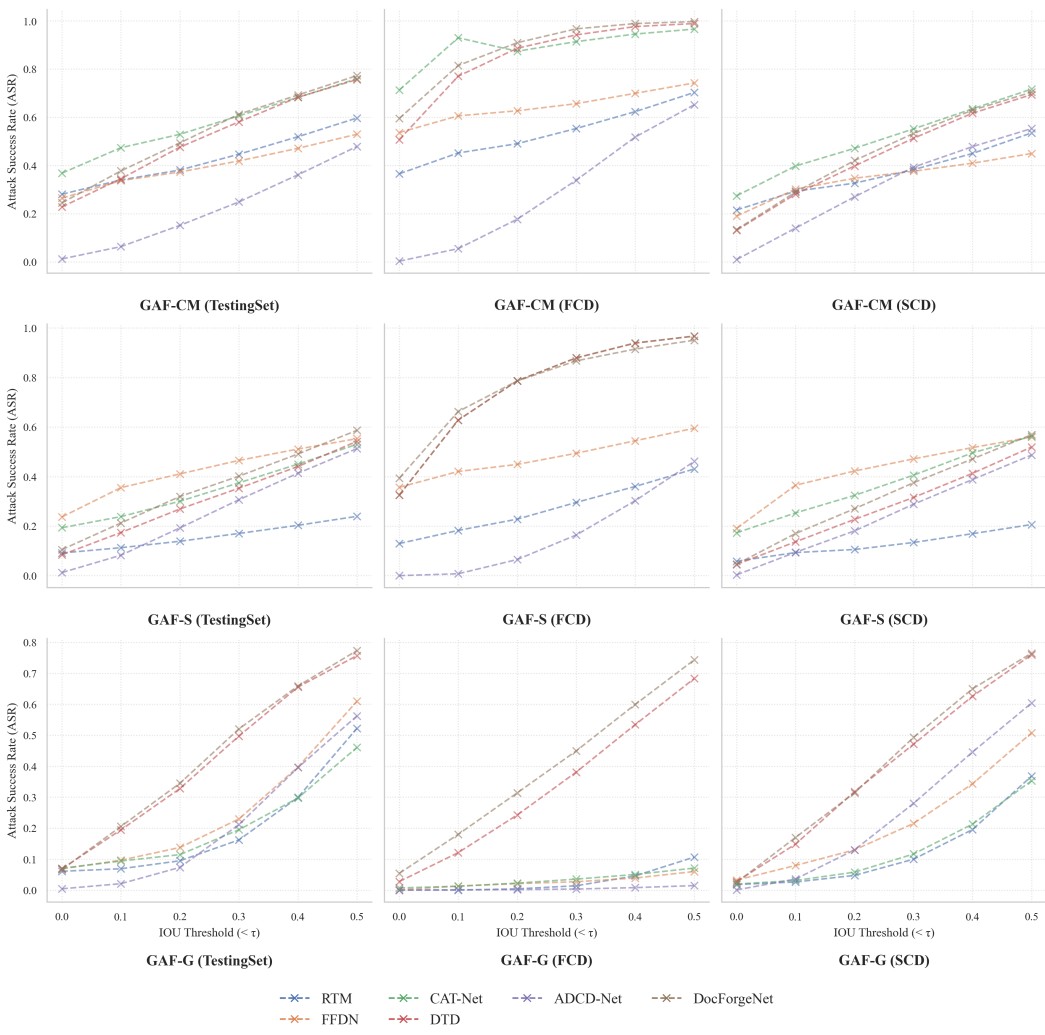

Figure 6: Attack Success Rate (ASR) vs. IOU threshold ($\tau$) under GAF attacks. Higher ASR indicates more successful evasion of detection. Results demonstrate that ASR increases with more lenient thresholds across all models, with GAF-CM and GAF-S proving substantially more effective than GAF-G, particularly on FCD split.

Finally, GAF-G attacks show a more gradual increase in ASR as the IOU threshold becomes more lenient, with most models maintaining strong performance under strict thresholds ($\tau < 0.1$). This pattern is particularly pronounced on the FCD split, where GAF-G proves notably less effective compared to other test splits: most models, except DTD and DocForgeNet, maintain robust performance even at $\tau = 0.5$. This aligns with our findings in Section 5.2, where GAF-G demonstrated substantially lower effectiveness compared to GAF-CM and GAF-S attacks.