# OpenReview forum: "THE JPEG BLIND SPOT: EXPOSING A CRITICAL VULNERABILITY IN DOCUMENT TAMPERING DETECTION"
_ICLR.cc/2026/Conference — Submitted to ICLR 2026_

### Official Review · Reviewer_o5xa · 2025-10-23

**Soundness:** 3
**Presentation:** 2
**Contribution:** 2
**Rating:** 4
**Confidence:** 5

**Summary:**

The paper exposes a major vulnerability in current document tampering detection models, such as DTD and DocForgeNet, which depend on frequency-domain cues like the JPEG compression–induced 8×8 block artifact grid (BAG). To exploit this weakness, the authors introduce two adversarial attack methods: (1) Grid-Aligned Copy-Move (GACM): This attack aligns tampered regions with existing 8×8 grids to preserve BAG consistency and evade detection. (2) Padding-Recompression-Cropping (PRC): This method intentionally offsets JPEG grids to produce artificial BAG inconsistencies, thereby triggering false positives. Extensive experiments on the DocTamper dataset reveal severe performance degradation under these attacks. For instance, the F1-score on the DocTamper-FCD test set drops dramatically from 76.2% to 3.1%. These results confirm that current detection systems largely rely on superficial frequency-domain artifacts rather than learning global semantic representations of tampering. The paper emphasizes the need for semantic-level detection models capable of robust and meaningful tampering analysis. It also highlights the security risks of the identified weakness in high-stakes applications and suggests that GACM and PRC can serve as benchmark attacks for evaluating the robustness of future detection models.

**Strengths:**

The experimental design demonstrates strong reliability and adaptability. Using DocTamper to assess performance in both standard and cross-domain contexts. By combining pixel-level Precision, Recall, and F1 metrics with qualitative visualizations, it ensures objective and intuitive evaluation. This framework offers a robust, reusable paradigm for future research.
The methodology is highly targeted and practical. The GACM and PRC attacks exploit the 8×8 JPEG block structure, exposing detection models’ overreliance on block-level artifacts instead of semantic features. Closely reflecting real-world tampering, these methods reveal key vulnerabilities and provide clear, reproducible insights for improving document forgery detection systems.

**Weaknesses:**

The core idea of this work is to exploit JPEG’s 8×8 block alignment to conceal tampering, a strategy that has already been investigated in prior natural image forensic research. The authors do not uncover a new vulnerability mechanism or make any fundamental theoretical contributions. As a result, the novelty lies more in practical instantiation or engineering implementation rather than in revealing the underlying essence of the problem or achieving a principled breakthrough.
The proposed attack is evaluated only on detectors that utilize a single DCT mode, the FPH module in DTD, within a two-stream architecture designed to exploit JPEG block artifacts. However, it does not assess detectors that employ alternative DCT configurations (such as those using dilated convolutions [1]) or multimodal architectures. Moreover, recent multi-branch approaches, such as Luo et al. [2], which integrate DCT, ELA, and noise features, are excluded from the evaluation. Consequently, the paper’s findings highlight vulnerabilities in a limited class of detectors under specific conditions, rather than demonstrating a general weakness across all DCT-based detection methods.
The authors do not discuss how training data or training strategies influence vulnerability to attacks. In our experiments with DTD, we observed that the DCT branch was susceptible to overfitting. However, applying data augmentations such as random image shifts effectively alleviated this problem while maintaining strong performance. If appropriate data augmentations or training strategies can mitigate attack impact, the practical novelty and risk of the proposed method would be reduced. The paper should therefore include experiments or at least a discussion addressing these factors.
Finally, the experimental design and visualization are inadequate. It only reports pre- and post-attack metrics without broader comparisons, ablations, or visual analyses to explain the attack mechanism or concealment of tampering traces. Moreover, it does not propose potential improvements to detection methods. For instance, whether DCT remains an effective feature for document-image forensics or how the DCT branch could be enhanced to resist such attacks. These are important aspects that should be discussed as directions for future research.

[1] Kwon M J, Yu I J, Nam S H, et al. CAT-Net: Compression artifact tracing network for detection and localization of image splicing[C]//Proceedings of the IEEE/CVF winter conference on applications of computer vision. 2021: 375-384.
[2] Luo D, Liu Y, Yang R, et al. Toward real text manipulation detection: New dataset and new solution[J]. Pattern Recognition, 2025, 157: 110828.

**Questions:**

Please clarify the novel contributions of this work beyond existing methods that leverage JPEG 8×8 alignment. Evaluate the proposed attack across a wider range of models, including those using DCT configurations with dilated convolutions, CAT-Net approaches, and multi-branch detectors such as Luo et al. Additionally, conduct experiments on training strategies and data augmentations (e.g., random pixel shifts) to determine whether these techniques can mitigate the attack. Include ablation studies and qualitative visualizations to illustrate the concealment mechanism, and propose or assess potential defenses or methodological improvements.

---

> ### Author Response · Authors · 2025-11-28
> **Official Comment by Authors**
>
> We thank the reviewers for their valuable feedback, and we have incorporated the requested changes into the revised manuscript accordingly.
>
> ## 1. Novelty of the core idea
> We fully agree that the concept of JPEG alignment is a well-known phenomenon in natural-image forensics. However, document images differ fundamentally: their glyph-like structures enable perfect alignment, intentional and trivial, not incidental as commonly reported for natural images. No prior document-forensics work formalizes or evaluates this as an adversarial vulnerability explicitly for modern Deep Learning-based detectors. We now explicitly highlight these points in our revised abstract, introduction, and the related work sections.
>
> ## 2. Model diversity and training strategies
> We have expanded our methodology and experiments to include **Splicing and Generative Forgeries**. See sections 4 and 5.2 for our additional experimental details and results. We introduce the GACM attack as GAF (Grid-Align Forgery) attack with subtypes GAF-CM (copy-move), GAF-S (splicing), and GAF-G (generative).
>
>
>    - Splicing forgeries: GAF applies directly, with similar degradation.
>    - Generative forgeries: Variable JPEG statistics make these easier to detect, but the vulnerability analysis remains valid.
>    - 3 additional models: ADCD-Net, RTM (AscFormer), and CatNet which are all state-of-the-art forgery methods. These model also show degradation under our attacks.
>
> All our results are now presented in The extended table 1 in main paper. Results on CAT-Net are shown below.
>
> | Model   | Attack    | Forgery    | F1 (TestingSet) | ASR (TestingSet) | F1 (FCD) | ASR (FCD) | F1 (SCD) | ASR (SCD) |
> |---------|-----------|------------|-----------------|------------------|----------|-----------|----------|-----------|
> | CAT-Net | No Attack | DocTamper  | 0.750           | -                | 0.937    | -         | 0.652    | -         |
> |         | PRC       |            | 0.624           | 0.011            | 0.595    | 0.042     | 0.546    | 0.009     |
> |         | No Attack | Copy-Move  | 0.781           | 0.230            | 0.953    | 0.116     | 0.795    | 0.148     |
> |         | GAF-CM    |            | 0.633           | 0.570            | 0.549    | 0.890     | 0.594    | 0.508     |
> |         | No Attack | Generative | 0.798           | 0.153            | 0.874    | 0.007     | 0.774    | 0.083     |
> |         | GAF-G     |            | 0.768           | 0.205            | 0.862    | 0.033     | 0.744    | 0.132     |
> |         | No Attack | Splicing   | 0.937           | 0.151            | 0.941    | 0.114     | 0.814    | 0.112     |
> |         | GAF-S     |            | 0.739           | 0.348            | 0.142    | 0.754     | 0.664    | 0.369     |
>
> Due to time constraints and complexity of re-training multiple models under identical settings, we were unable to include such experiments in this revision. However, we now explicitly also discuss the robustness of different models against our attacks in the paper. For instance, for FFDN we perform an investigation into which of its modules is mainly responsible for its higher robustness compared to DTD or DocForgeNet models. These results are reported in Appendix B. Similarly, our additional experimental evaluation of the ASR metric under varying thresholds in Appendix C highlights the higher robustness of ADCD-Net and RTM models against our attacks relative to other models.

---

### Official Review · Reviewer_1DEi · 2025-10-28

**Soundness:** 3
**Presentation:** 2
**Contribution:** 2
**Rating:** 4
**Confidence:** 5

**Summary:**

This paper exposes a critical vulnerability in state-of-the-art document tampering detection models, which the authors argue rely excessively on low-level JPEG compression artifacts—specifically Block Artifact Grids (BAG)—rather than learning semantically meaningful representations of forgery. To demonstrate this, the authors introduce two novel adversarial attacks: a Grid-Aligned Copy-Move (GACM) attack that preserves local JPEG block statistics during tampering, and a Pad-Recompress-Crop (PRC) attack that deliberately shifts the JPEG grid to induce false positives. When evaluated on the DocTamper benchmark, these attacks catastrophically reduce the performance of leading detectors—sometimes to near-random levels—revealing their fundamental reliance on superficial artifact cues and underscoring the need for more robust, semantically-aware forensic systems.

**Strengths:**

1. Originality:
While the existence of JPEG grid artifacts and their use in forensics is a well-established field, the core insight—that state-of-the-art (SotA) deep learning models have a critical, over-reliant, and exploitable "blind spot" centered on these artifacts—is novel. The paper models expose a fundamental flaw of the DCT-based forensic models.
2. Quality:
The methodology is sound and well-grounded in the JPEG compression standard. The experimental setup is thorough: It tests against multiple, recent SotA models (DTD, DocForgeNet, FFDN). It employs a realistic attack implementation using OCR for text box selection. It provides both quantitative results (showing catastrophic drops in F1-score to as low as 1-3%) and qualitative visualizations that make the failure modes immediately understandable. The fact that one model (FFDN) shows greater robustness is noted and plausibly explained (its use of a Visual Enhancement Module), which actually strengthens the paper's argument by showing that reducing reliance on frequency-domain features improves resilience.
3. Clarity:
The paper is well-written and structured. It efficiently establishes the problem context, clearly states its central hypothesis, and provides an intuitive explanation of the JPEG artifact vulnerability before delving into technical details. The attacks (GACM and PRC) are described with precise mathematical formulations and accompanied by clear pseudocode and illustrative figures. The results are presented in a comprehensive table and supported by qualitative figures that effectively demonstrate both the "false negative" (GACM) and "false positive" (PRC) failure modes.

**Weaknesses:**

1. Scope and Generality of the Attacks:
The paper's central claim is that it exposes a "critical vulnerability in document tampering detection." However, the attacks are demonstrated primarily against a specific type of forgery: copy-move. Document tampering also includes splicing (inserting elements from a different image) and generative forgeries (using AI to alter text in-place).
2. Depth of Evaluation Metrics:
The evaluation relies exclusively on pixel-level Precision, Recall, and F1-score. While standard, these metrics do not fully capture the "catastrophic failure" in a security context. Suggestions: For GACM (False Negatives): Report the Attack Success Rate (ASR), i.e., the percentage of forged instances where the model's detection score in the tampered region falls below a operational threshold (e.g., where the F1-score drops to near zero). The current F1-score is an aggregate; an ASR would more starkly show how often the attack completely bypasses detection. For PRC (False Positives): Quantify the "unreliability" by reporting the False Positive Area (FPA) or the percentage of the original, authentic image that is now flagged as tampered after the PRC attack. A large FPA would powerfully demonstrate how the model becomes practically unusable. Segment-level Evaluation: Pixel-level F1 can be harsh. A segment-level metric (e.g., IoU of connected components) could show that the model not only performs poorly pixel-wise, but fails to localize the entire forged text segment, which is often the operational goal.
3. Analysis of Defenses and Robust Representations:
The paper effectively critiques existing models but offers limited guidance on how to build robust ones. The observation that FFDN is more robust is a starting point, but the analysis is superficial.

**Questions:**

1. On the Scope and Generality of the Vulnerability:
1.1 The demonstrated attacks focus exclusively on copy-move forgeries. Is the identified vulnerability fundamental to the models' architecture, and therefore also applicable to other common document forgery types, such as splicing (inserting text from a different image) or **generative in-painting**? Could you demonstrate a "Grid-Aligned Splice" attack or discuss why the same principles would/would not apply?
1.2 The attacks exploit the JPEG 8x8 grid. How does the vulnerability manifest with other common document image formats, such as PNG (lossless) or JPEGs with different block sizes? If a document is stored in a lossless format, do these models fail completely, or do they fall back to other (potentially more semantic) features?

2. On the Evaluation and Metrics:
2.1 The current pixel-wise F1-score, while standard, does not fully capture the security implications of a failed detection. We suggest reporting additional metrics: Attack Success Rate (ASR): For GACM, what percentage of forged instances are completely missed (e.g., IoU = 0 or F1 < 0.1)? This directly measures how often the attack bypasses the detector entirely. False Positive Area (FPA): For PRC, what percentage of the original, authentic image is falsely flagged as tampered? This quantifies the "systematic false positives" and the resulting unreliability.
2.2 The PRC attack induces false positives, but are these false positives random noise, or do they correlate with specific semantic structures in the document (e.g., text glyph edges, line intersections)? A qualitative analysis of where the false positives occur could provide deeper insight into the model's faulty reasoning process.

3. On the Architectural Analysis and Pathways to Robustness:
3.1 You hypothesize that FFDN's relative robustness is due to its Visual Enhancement Module (VFM) reducing reliance on frequency features. Did you perform any ablation studies to confirm this? For instance, if you remove the VFM from FFDN, does its performance on GACM drop to the level of DTD? Conversely, if you add a similar RGB-fusion module to DTD, does its robustness improve?
3.2 Have you explored simple defensive strategies or data augmentations based on your findings? For example, does training a model on a dataset that includes examples with random PRC-style grid shifts improve its robustness to both GACM and PRC attacks without harming its performance on standard forgeries?

---

> ### Author Response · Authors · 2025-11-28
> **Official Comment by Authors**
>
> We thank the reviewer for the valuable feedback. We have incorporated the suggested changes in the paper now.
>
> **1. Applicability beyond copy-move**
>
> We have expanded our methodology and experiments to include **Splicing and Generative Forgeries**. See sections 4 and 5.2 for our additional experimental details and results. We introduce the GACM attack as GAF (Grid-Align Forgery) attack with subtypes GAF-CM (copy-move), GAF-S (splicing), and GAF-G (generative).
>
>
> Splicing forgeries: GAF-S applies directly, with similar degradation.
> Generative forgeries: Variable JPEG statistics make these easier to detect, but the vulnerability analysis remains valid.
>
> All our results are now presented in The extended table 1.
>
>
> **2. Additional metrics**
>
> Following your suggestions, we now explicitly report **Attack Success Rate (ASR)** alongside **False positive Area (FPA)** in the main text. We define ASR as the total number of images on which the intersection-over-union (IoU) between the ground-truth and the predicted pixels is less than a target threshold and report the average ASR over multiple thresholds. **False Positive Area (FPA)** metric computes the fraction of pixels that are incorrectly predicted as tampered by the model and is reported average over all samples. See Section 5. Attack Evaluation Metrics for more details. We also report ASR/FPA now in Table 1. and Table 2 (in Appendix B) in Section 5.2. Finally we also provide ASR vs threshold experimentation results in Appendix C to evaluate how well each model performs from strict to lenient threshold regimes (0.0 to 0.5).
>
>
> **3. Architectural analysis**
>
> In our new ablation experiments in Appendix B, we explicitly investigate the FFDN module. Since FFDN builds upon the DTD (Qu et al.,2023) architecture, with the addition of Vision Enhancement Module (VEM) and WFE module to strengthen its feature representation and cross-modal integration, in our ablation, we evaluate the model with and without its WFE module and show that its performance difference remains negligible under both setups. This leaves the VEM module to be the only remaining addition over DTD that is responsible for its better robustness. This is also expected as VEM is explicitly designed for better fusion of DCT and RGB features as highlighted in their work. The ablation results are presented below:
>
> | Model        | Attack   | Forgery    | F1 (TestingSet) | ASR/FPA (TestingSet) | F1 (FCD) | ASR/FPA (FCD) | F1 (SCD) | ASR/FPA (SCD) |
> |--------------|----------|------------|----------------|----------------------|---------|---------------|---------|---------------|
> | FFDN w/ WFE  | No Attack| DocTamper  | 0.956          | -                    | 0.916   | -             | 0.812   | -             |
> |              | PRC      |            | 0.752          | 0.001                | 0.710   | 0.002         | 0.735   | 0.001         |
> |              | No Attack| Copy-Move  | 0.815          | 0.112                | 0.915   | 0.024         | 0.876   | 0.062         |
> |              | GAF-CM   |            | 0.549          | 0.399                | 0.314   | 0.645         | 0.613   | 0.346         |
> |              | No Attack| Generative | 0.693          | 0.185                | 0.946   | 0.004         | 0.736   | 0.122         |
> |              | GAF-G    |            | 0.621          | 0.257                | 0.918   | 0.027         | 0.650   | 0.218         |
> |              | No Attack| Splicing   | 0.723          | 0.247                | 0.922   | 0.018         | 0.766   | 0.194         |
> |              | GAF-S    |            | 0.552          | 0.422                | 0.475   | 0.477         | 0.550   | 0.421         |
> | FFDN w/o WFE | No Attack| DocTamper  | 0.953          | -                    | 0.915   | -             | 0.808   | -             |
> |              | PRC      |            | 0.747          | 0.001                | 0.705   | 0.002         | 0.729   | 0.001         |
> |              | GUCM     | Copy-Move  | 0.809          | 0.118                | 0.915   | 0.024         | 0.868   | 0.074         |
> |              | GAF-CM   |            | 0.534          | 0.417                | 0.303   | 0.657         | 0.593   | 0.369         |
> |              | GUG      | Generative | 0.695          | 0.193                | 0.944   | 0.004         | 0.724   | 0.137         |
> |              | GAF-G    |            | 0.609          | 0.272                | 0.910   | 0.033         | 0.632   | 0.240         |
> |              | GUS      | Splicing   | 0.695          | 0.288                | 0.922   | 0.020         | 0.734   | 0.228         |
> |              | GAF-S    |            | 0.511          | 0.465                | 0.467   | 0.482         | 0.512   | 0.462         |

---

### Official Review · Reviewer_Ap6R · 2025-10-30

**Soundness:** 2
**Presentation:** 2
**Contribution:** 1
**Rating:** 2
**Confidence:** 4

**Summary:**

This paper identifies a critical vulnerability in current state-of-the-art document tampering detection models, which primarily rely on JPEG compression artifacts, specifically Block Artifact Grids (BAGs), for localizing forged regions. The authors propose two adversarial attacks: Grid-Aligned Copy–Move (GACM) and Pad–Recompress–Crop (PRC). GACM preserves local JPEG block statistics, making forged regions consistent with the 8x8 grid, while PRC deliberately shifts the JPEG grid to induce widespread false positives. Evaluating these attacks on the DocTamper benchmark against leading detection methods (DTD, DocForgeNet, FFDN), the study demonstrates a catastrophic failure, reducing detection rates to near-random chance or triggering significant false positives. This work highlights that existing models often fail to learn genuine semantic tampering representations, instead relying on superficial compression artifacts, thus underscoring a need for more robust forensic systems.

**Strengths:**

The paper introduces PRC, specifically designed to exploit the underlying mechanism of JPEG compression artifacts. This directly challenges the current paradigm of frequency-domain feature reliance. PRC deliberately misaligns the JPEG grid without visible changes to the document, effectively turning benign images into "tampered" ones from the perspective of current detectors. This highlights a severe blind spot in existing systems.

The study benchmarks against three leading state-of-the-art models (DTD, DocForgeNet, FFDN), all of which explicitly leverage frequency-domain DCT features. This direct comparison against strong baselines validates the effectiveness of the proposed attacks.

**Weaknesses:**

On Grid-Aligned Copy-Move (GACM) and Novelty:
The observation that Grid-Aligned Copy-Move (GACM) can avoid Block Artifact Grids (BAGs) is a well-established concept in the field of JPEG image forensics. Prior literature has extensively documented this phenomenon, and it's recognized that even random copy-move operations have a 1/64 chance of aligning with grid boundaries. Consequently, the discussion of GACM in this paper does not represent a novel contribution.

On Grid Shift via Pad Recompress Crop (PRC) and Comparative Strength:
While Grid Shift via Pad Recompress Crop (PRC) is a recognized form of image distortion, other common image distortions such as iteratively resize and compress can also destroy the BAGs and leads to performance degradation. It is essential to explicitly articulate the unique strengths or advantages of PRC in degrading forensic performance when compared to these alternative, potentially equally effective, common distortion techniques.

Limited Scope of the Adversarial Attack:
The adversarial attack presented appears to have a significantly limited scope. Firstly, it is stated to apply primarily to frequency-based forensic models, with its impact on non-frequency-based forensic models being negligible. Secondly, its applicability is restricted solely to copy-move forgeries. There is no discussion or demonstration of its extensibility to other prevalent forgery methods, such as splicing or those involving AI-generated content.

Lack of Evaluation Against Robust Models:
A significant omission in the current work is the lack of evaluation against contemporary forensic models specifically designed for robustness against adversarial attacks, such as ADCD-Net [1]. Without experiments or analysis demonstrating the proposed method's efficacy against such specialized defenses, its overall impact and practical relevance in a robust forensic landscape remain unquantified.

[1] Wong, Kahim et al. ADCD-Net: Robust Document Image Forgery Localization via Adaptive DCT Feature and Hierarchical Content Disentanglement. ICCV2025.

**Questions:**

No other questions.

---

> ### Author Response · Authors · 2025-11-28
> **Official Comment by Authors**
>
> We thank the reviewer for their valuable feedback.
>
> ## 1. Novelty of GACM
> We fully agree that the concept of JPEG alignment is a well-known phenomenon in natural-image forensics. However, document images differ fundamentally: their glyph-like structures enable perfect alignment, intentional and trivial, not incidental as commonly reported for natural images. No prior document-forensics work formalizes or evaluates this as an adversarial vulnerability explicitly for modern Deep Learning-based detectors. **We now explicitly highlight these points in our revised abstract, introduction, and the related work sections.**
>
> ## 2. Strength of PRC vs. other perturbations
> Other types of destructive attacks such as iterative resize and compress cycles would also destroy the RGB information whereas our PRC attack only shifts the DCT grid while preserving the image space features before and after the attack. This ensures that the DCT structure undergoes only a controlled, origin preserving shift and so we can explicitly evaluate the overreliance of the model on the DCT-space biases.
>
> ## 3. Scope beyond copy-move
> We have expanded our methodology and experiments to include **Splicing and Generative Forgeries**. See sections 4 and 5.2 for our additional experimental details and results. We introduce the GACM attack as GAF (Grid-Align Forgery) attack with subtypes GAF-CM (copy-move), GAF-S (splicing), and GAF-G (generative).
>
>
> Splicing forgeries: GAF-S applies directly, with similar degradation.
> Generative forgeries: Variable JPEG statistics make these easier to detect, but the vulnerability analysis remains valid.
>
> All our results are now presented in The extended table 1.
>
> ## 4. Evaluation on robust models
> We have now added 3 additional models: ADCD-Net and RTM (Luo et al.) and Catnet (Kwon et al.) improving the generalization of our work (refer to table 1). ADCD-Net shows similar GACM failure rates, but shows more robustness overall compared to other models. See also additional results on in Appendix C where we highlight the newly added Attack Success Rate (ASR) metric. Few results of the newly added models are shown below (please see Table 1 in paper for completeness).
>
> | Model      | Attack   | Forgery     | F1 (TestingSet) | ASR (TestingSet) | F1 (FCD) | ASR (FCD) | F1 (SCD) | ASR (SCD) |
> |-----------|----------|-------------|----------------|----------------------|---------|---------------|---------|---------------|
> | CAT-Net   | No Attack| Copy-Move   | 0.781          | 0.230                | 0.953   | 0.116         | 0.795   | 0.148         |
> |           | GAF-CM   |             | 0.633          | 0.570                | 0.549   | 0.890         | 0.594   | 0.508         |
> |           | No Attack| Generative  | 0.798          | 0.153                | 0.874   | 0.007         | 0.774   | 0.083         |
> |           | GAF-G    |             | 0.768          | 0.205                | 0.862   | 0.033         | 0.744   | 0.132         |
> |           | No Attack| Splicing    | 0.937          | 0.151                | 0.941   | 0.114         | 0.814   | 0.112         |
> |           | GAF-S    |             | 0.739          | 0.348                | 0.142   | 0.754         | 0.664   | 0.369         |
> | RTM       | No Attack| Copy-Move   | 0.657          | 0.251                | 0.698   | 0.228         | 0.734   | 0.173         |
> |           | GAF-CM   |             | 0.498          | 0.428                | 0.398   | 0.531         | 0.556   | 0.367         |
> |           | No Attack| Generative  | 0.699          | 0.157                | 0.867   | 0.016         | 0.762   | 0.079         |
> |           | GAF-G    |             | 0.649          | 0.201                | 0.933   | 0.028         | 0.708   | 0.126         |
> |           | No Attack| Splicing    | 0.793          | 0.146                | 0.781   | 0.117         | 0.823   | 0.108         |
> |           | GAF-S    |             | 0.776          | 0.159                | 0.627   | 0.271         | 0.805   | 0.128         |
> | ADCD-Net  | No Attack| Copy-Move   | 0.618          | 0.154                | 0.757   | 0.115         | 0.698   | 0.221         |
> |           | GAF-CM   |             | 0.532          | 0.219                | 0.436   | 0.291         | 0.607   | 0.307         |
> |           | No Attack| Generative  | 0.685          | 0.137                | 0.970   | 0.041         | 0.730   | 0.128         |
> |           | GAF-G    |             | 0.577          | 0.211                | 0.940   | 0.005         | 0.575   | 0.249         |
> |           | No Attack| Splicing    | 0.632          | 0.183                | 0.802   | 0.075         | 0.707   | 0.178         |
> |           | GAF-S    |             | 0.547          | 0.253                | 0.618   | 0.167         | 0.639   | 0.240         |

---

### Official Review · Reviewer_ETGT · 2025-11-01

**Soundness:** 2
**Presentation:** 2
**Contribution:** 2
**Rating:** 2
**Confidence:** 5

**Summary:**

This paper exposes the vulnerability inherent in BAG-based document tampering localization. In details, they propose two attack methods to fool existing methods; and evaluate the proposed attacks on benchmark datasets.

**Strengths:**

1. The authors have a thorough understanding of existing BAG-based vanilla document tamper detection methods, which is beneficial and necessary for attacking.

2. The authors clarify the mechanism of attacks clearly.

**Weaknesses:**

1. The stated conclusion for the failed defense against the proposed attacks are flawed. For hand-crafted tampering, such as copy-move, the tampered regions are not required to be generated. For the original reference image, the regions labeled as tampering are intrinsically authentic, but their designation stems solely from their anomalous location. Therefore, forensic models to solve hand-crafted forgeries can not extract genuine semantic representations of tampering patterns, which is different from DeepFake detection; what they can make use of is the inconsistencies between the labeled authentic and tampered regions in low-level patterns, such as BAG. If the statistics of such low-level patterns change, it is reasonable and natural that the performance declines. Additionally, if you change the format of targeted images for tampering localization, such as, “.png”, maybe all BAG-based methods can not work. There is no need to extensively design attack methods.

2. The GACM has limitations. The precise grid distribution must be known. For example, if the target image has been cropped and its origin of DCT grid is not the top-left corner, how to determine the attack area's position is a major challenge.

3. Experimental results do not always reflect the effectiveness of the attack methods. For example, the PRC attack is not significant for the forensic model, FFDN.

**Questions:**

none

---

> ### Author Response · Authors · 2025-11-28
> **Official Comment by Authors**
>
> Thank you for your valuable and constructive feedback. We appreciate the time and effort you invested in reviewing our work, and we have incorporated all relevant suggestions and revisions into the updated version of the paper.
>
> ## 1. Interpretation of model behavior on handcrafted forgeries
>
> We agree that current detectors heavily rely on low level BAG and DCT cues; our work precisely examines the consequences of this dependence.
> The study remains important because:
>
> - **Existing Models claim cross forgery generalization:**
>   Many evaluated methods state that as long as RGB and DCT inputs are available, performance generalizes across tampering types. Under this premise, we believe it is reasonable to evaluate them in realistic scenarios where a forger may deliberately minimize inconsistencies in the DCT space. We have revised the paper introduction, and related work to further highlight these points.
>
> - **DCT overreliance is empirically demonstrated:**
>   Qualitative results show misclassifications even when tampering is obvious visually, contradicting robustness claims. We have highlighted this explicitly in Section 5.3 of the paper in Fig.4 and Fig 5.
>
> - **We provide a practical audit mechanism:**
>   GAF/PRC act as lightweight diagnostics to ensure future models fuse RGB+DCT meaningfully rather than depending on frequency artifacts.
>
> ## 2. Changing image format (.png)
>
> (1) If a tampered image in JPEG is converted to PNG, it does not prevent BAG based methods because the original JPEG grid structure remains preserved and DCT can be recomputed from the pixel space.
> (2) If we are only dealing with PNG images, then no BAG-based methods or our attacks are applicable, but that is a different problem domain.
>
> **JPEG remains our focus because:**
>
> - Most document forensic systems operate on JPEG.
> - Multiple recompressions reduce detectability (Doctamper).
>
> We also now discuss this point in the newly added **Section 6. Limitations** in the main text of our revised manuscript.
>
>
>
> ## 3. GACM requires grid origin knowledge
>
> This is an excellent point! However, we would like to point out that instead of a limitation, it is more of a basic assumption for GAF (previously GACM) to work. For GAF (previously GACM) attack, it is natural to assume the initial image that undergoes forgery is not already forged/edited just as this same assumption applies to the original setting in which the model is evaluated (where its grid origin remains unshifted). Because if the image is cropped and the origin of its DCT grid is shifted, this becomes equivalent to our PRC attack which we already propose in parallel to cater for this scenario. We now discuss this point in the newly added Section 6. Limitations in the main text of our revised manuscript.
>
> Furthermore, we would like to mention that while we propose the GAF attacks as one systematic exploit and it may not work under all types of scenarios, we believe we are still the first to provide a structured approach for revealing the over-reliance of state-of-the-art document forgery detectors on DCT-domain cues. Even if our attacks fail under certain assumptions, they can serve as an auditing tool for validating that a deep learning-based model is not heavily biased towards the frequency domain features.
>
>
> ## 4. PRC has limited effect on FFDN
>
> We agree that experimental results do not always reflect the effectiveness of the attack methods. However, our detailed evaluation shows that PRC exposes two distinct failure modes across detectors. Models such as DTD and DocForgeNet experience severe false-positive inflation under PRC (e.g., FPA 0.177 and 0.215), often marking large regions of clean text as tampered. FFDN does not exhibit this extreme behavior (Fig. 5), but PRC still disrupts its performance in a meaningful way. Specifically, FFDN’s F1-score drops from 0.956 to 0.752 (Table. 1), even though the RGB content remains unchanged and the attack only shifts JPEG grids. This decline confirms that PRC interferes with FFDN’s DCT alignment rather than triggering the large-scale over-detection seen in other models. Our ablation study (Appendix B) shows that the Vision Enhancement Module (VEM) improves FFDN’s RGB–DCT fusion, which explains its relative stability; however, the VEM only reduces, not removes, PRC sensitivity. Thus, PRC reveals a complementary vulnerability: FFDN avoids false-positive explosions but still suffers a 20–25% performance drop due to grid-shift–induced feature misalignment. Below are the results for FFDN.
>
> | Model | Attack | F1 Test | FPA Test | F1 FCD | FPA FCD | F1 SCD | FPA SCD |
> |-------|--------|---------|---------------|--------|--------------|--------|--------------|
> | FFDN  | No Attack | 0.956 | -     | 0.916 | -     | 0.812 | -     |
> | FFDN  | PRC       | 0.752 | 0.001 | 0.710 | 0.002 | 0.735 | 0.001 |

---

### Author Response · Authors · 2025-11-28
**Official Meta Statement by Authors**

We thank the reviewers for their constructive feedback. While JPEG block alignment is a known phenomenon in natural image forensics, our work makes a distinct contribution by formalizing this phenomenon as an adversarial diagnostic tool in the document-image domain, where alignment is not accidental but trivial to construct. Our advances include:

- **Document-specific domain shift:** Unlike natural images (1/64 chance of incidental alignment), document images contain sharply bounded glyph regions that make Grid-Aligned Forgery (GAF, previously GACM) straightforward and realistic for real-world manipulation.

- **Adversarial audit formulation:** We are the first to formalize GAF and PRC as controlled adversarial attacks for multimodal RGB+DCT detectors, revealing systematic over-reliance on fragile DCT patterns rather than rediscovering JPEG alignment.

- **Beyond copy–move:** We show that GAF extends naturally to splicing (updated Table 1 in the main manuscript), exposing the same severe failure modes. The issue is architectural, not tied to a specific forgery type.

- **Evaluation on robustness-oriented models:** We include ADCD-Net and Luo et al. (RTM), showing that even robustness-focused architectures fail under GAF (Table 1).

- **Uniqueness of PRC:** Unlike generic distortions, PRC preserves RGB content and BAG structure while shifting only the grid origin, allowing us to isolate true cross-modal fragility.

- **New metrics:** We added Attack Success Rate (ASR) for GAF and False Positive Area (FPA) for PRC (Table 1) to highlight measurable, security-relevant failures as suggested by a reviewer.

- **Architectural ablation:** Our new ablation confirms that FFDN’s robustness stems solely from its Visual Enhancement Module (Appendix B), validating DTD ↔ FFDN as a clean architectural comparison.

---

### Author Response · Authors · 2025-12-03
**Comprehensive Response to Reviewer Feedback and Major Manuscript Revisions**

We sincerely thank the reviewers for their thoughtful and detailed feedback. In response, we have undertaken a major, comprehensive revision of the manuscript, expanding experiments, refining methodology, adding new analyses, incorporating new baselines, introducing new metrics, and clarifying assumptions. Nearly every section of the paper has been strengthened as a result.

## **Scope of Revisions (Significant Additions & Improvements)**

### **1. Substantial Conceptual Refinement**
We now clearly articulate why **GAF** and **PRC** represent *document-specific adversarial vulnerabilities* that do not exist in natural-image settings. This distinction is emphasized in the **Abstract**, **Introduction**, and **Related Work** sections.

### **2. Expanded Adversarial Framework**
We extended **GAF** into a full taxonomy:
- **GAF-CM** (Copy–Move)
- **GAF-S** (Splicing)
- **GAF-G** (Generative Forgeries)

with new experiments added for each category.

### **3. Broader and Deeper Experimental Evaluation**
We added **three new state-of-the-art models**:
- **ADCD-Net**
- **RTM / AscFormer**
- **CatNet**

We also significantly expanded **Table 1** and **Appendix C** with extensive cross-model comparisons and added robustness analyses.

### **4. New Evaluation Metrics (ASR + FPA)**
Following reviewer suggestions, we introduced:

- **Attack Success Rate (ASR)**
  $$\text{ASR} = \frac{ \text{ No. of successful adversarial attacks}}{\text{No. of total attempts}}$$

- **False Positive Area (FPA)**
  $$\text{FPA} = \frac{\text{Area incorrectly predicted as forged}}{\text{Total image area}}$$

These metrics enable a more rigorous and security-relevant assessment of model failure modes.

### **5. New Architectural Ablations**
A detailed ablation study (Appendix B) isolates the source of **FFDN**'s robustness, demonstrating that the **Visual Enhancement Module (VEM)** is the key contributor.

### **6. Clarified Assumptions & Limitations**
A newly added **Section 6** now explicitly covers:
- grid-origin assumptions,
- JPEG→PNG transformation issues,
- applicability boundaries of **GAF/PRC**.

### **7. Enhanced Qualitative Explanations**
We expanded qualitative examples (**Fig. 4 & 5**) to illustrate:
- DCT overreliance,
- visual inconsistencies,
- misclassification behavior.

---

## **Overall Effort**
We have made extensive and substantial revisions across the manuscript, addressing every reviewer concern in depth. The paper is now significantly clearer, more rigorous, and more comprehensive. We hope these major additions assist the Area Chair in making an informed decision.

---

### Meta-Review · Area_Chair_bstf · 2026-01-08

**Summary:**

The reviewers agree that this paper exposes a vulnerability in current document forgery detection models due to over-reliance on JPEG Block Artifact Grids (BAGs), and the proposed GAF and PRC attacks demonstrate substantial performance degradation across several detectors. The rebuttal and revisions have strengthened the work: attacks are extended beyond copy-move to splicing and generative forgeries, additional robust models are evaluated, new metrics (ASR and FPA) are introduced, and architectural ablations clarify sources of robustness. Despite these improvements, the core novelty remains largely practical or engineering-focused rather than theoretical, and the attacks are limited to JPEG-based detectors, leaving questions about generality to multimodal architectures or non-JPEG formats.

My recommendation is borderline toward reject. While the paper provides a structured evaluation framework and useful benchmarks, it does not advance fundamental understanding of adversarial robustness or detection principles to the level expected for acceptance.

**Reviewer Concerns:**

The rebuttal has addressed several concerns: scope beyond copy-move (Reviewers 2, 3, 4), evaluation on robust models: added ADCD-Net, RTM/AscFormer, CatNet (Reviewers 2, 4).

For Reviewer 3's concerns, rebuttal provided additional metrics: including Attack Success Rate (ASR) and False Positive Area (FPA). Regarding architectural analysis, ablations showing FFDN robustness stems from VEM.

There are still outstanding concerns not addressable by rebuttal (i.e. not ready in this version of submission): The core idea largely engineering/practical rather than theoretical, as pointed out by Reviewers 2 and 4. The attacks remain JPEG-focused and unclear applicability to multimodal detectors or non-JPEG formats. The proposed method still does not advance theory of robust detection or adversarial defenses.

**Reviewer Scores:**

Reviewer 1 and 2 would likely remain scores of 2 or increased to 3.

Reviewer 3 and 4 has original scores of 4, and would remain.

---

### Decision · Program_Chairs · 2026-01-26

Reject